# Transmembrane but not soluble helices fold inside the ribosome tunnel

Manuel Bañó-Polo[1], Carlos Baeza-Delgado[1], Silvia Tamborero[1], Anthony Hazel[2], Brayan Grau[1], IngMarie Nilsson[3], Paul Whitley[4], James C. Gumbart[2], Gunnar von Heijne [3] & Ismael Mingarro [1]

Integral membrane proteins are assembled into the ER membrane via a continuous ribosome-translocon channel. The hydrophobicity and thickness of the core of the membrane bilayer leads to the expectation that transmembrane (TM) segments minimize the cost of harbouring polar polypeptide backbones by adopting a regular pattern of hydrogen bonds to form α-helices before integration. Co-translational folding of nascent chains into an α-helical conformation in the ribosomal tunnel has been demonstrated previously, but the features governing this folding are not well understood. In particular, little is known about what features influence the propensity to acquire α-helical structure in the ribosome. Using in vitro translation of truncated nascent chains trapped within the ribosome tunnel and molecular dynamics simulations, we show that folding in the ribosome is attained for TM helices but not for soluble helices, presumably facilitating SRP (signal recognition particle) recognition and/or a favourable conformation for membrane integration upon translocon entry.

[1] Estructura de Recerca Interdisciplinar en Biotecnologia i Biomedicina (ERI BioTecMed), Departament de Bioquímica i Biologia Molecular, Universitat de València, E-46100 Burjassot, Spain. [2] School of Physics, School of Chemistry and Biochemistry, Parker H. Petit Institute for Bioengineering and Bioscience, Georgia Institute of Technology, Atlanta, GA, USA. [3] Center for Biomembrane Research, Department of Biochemistry and Biophysics, Stockholm University, SE-10691 Stockholm, Sweden. [4] Department of Biology and Biochemistry, Centre for Regenerative Medicine, University of Bath, Bath BA2 7AY, UK. These authors contributed equally: Manuel Bañó-Polo, Carlos Baeza-Delgado, Silvia Tamborero, Anthony Hazel, Brayan Grau. Correspondence and requests for materials should be addressed to I.M. (email: Ismael.Mingarro@uv.es)

Membrane-spanning domains of integral membrane proteins must achieve their final folded structure in a very different environment, the hydrophobic interior of a lipid bilayer, compared to that experienced by soluble proteins. In the membrane environment, there is a strong driving force for polypeptide chains to adopt regular secondary structure (mainly α-helical) in order to reduce the significant free energy penalty of exposing polar peptide bonds to the hydrophobic core of biological membranes. Thus, the formation of α-helices, stabilized by a regular hydrogen bonding network of polar peptide bonds, is essential for the folding and insertion of transmembrane (TM) segments into biological membranes.

In the biogenesis of all proteins the nascent polypeptide must navigate through the ribosomal tunnel toward the exit site. For the vast majority of eukaryotic integral membrane proteins, nascent chains are elongated by ribosomes following targeting of a translationally stalled ribosome/nascent chain/SRP complex to the translocation/insertion machinery, i.e. the Sec61 translocon in the ER membrane. The translocon facilitates the insertion of TM segments into the lipid bilayer[1] in addition to the translocation of lumenal regions of membrane proteins and secreted proteins across the ER membrane[2,3]. The alignment of the ribosome exit site with the central pore of the translocon is proposed to facilitate direct movement of the elongating polypeptide from the ribosomal exit tunnel across or into the membrane[4]. The internal diameters of both the ribosomal exit tunnel[5] and the translocon[6] range from ~10 to 20 Å[7,8], which have been shown to be sufficient to allow secondary structure formation of α-helices in elongating nascent polypeptide chains[9–12].

With the importance of co-translational acquisition of structure while the nascent polypeptide chain is still tethered at the ribosomes being well-established[13], and folding of tethered nascent chains into an α-helical conformation in the ribosomal tunnel demonstrated[9–11,14–16], it is unclear what features of a helical region influence the propensity to acquire an α-helical structure whilst still in the ribosome. In particular, given that a TM segment should be folded prior to its exposure to the lipidic environment for thermodynamic reasons[17,18], we considered that α-helical TM segments might achieve secondary structure in a different location/environment than helices in water-soluble proteins.

To address this possibility, we used truncated nascent chains trapped within the ribosome-translocon complex of a model protein (*E. coli* leader peptidase (Lep)) containing engineered 'test' sequences of amino acids with known helical propensity in their final folded forms. Whereas these test sequences had different biophysical properties, i.e. were hydrophobic TM stretches of amino acids or hydrophilic non-TM (soluble) sequences, they were of similar length. We measure the number of residues of nascent polypeptide (*d*, distance P-NST) required to span the distance between the P-site on the ribosome (located at the entrance to the ribosomal tunnel) and the active site of the oligosaccharyl transferase (OST) (located nearby the lumenal end of the translocon central pore)[19,20]. By translating truncated nascent polypeptide chains of different lengths we observe that test sequences containing TM sequences required a larger number of residues to reach the glycosylation acceptor site in comparison to non-TM sequences. This suggests a more compact conformation for nascent polypeptides harboring TM helices, indicating that these helices are formed prior to exit from the ribosome and consequently prior to integration into the lipid bilayer from the translocon pore. By the same token, non-TM stretches do not attain full helical conformations far inside the ribosome tunnel. Molecular dynamics simulations of the helical test sequences in folded and extended states inside the ribosome exit tunnel were also performed. The simulations reveal that folding in the ribosome is favorable for TM helices, but unfavorable for soluble helices. The study also demonstrates that measured TM helix folding efficiencies are dependent on whether the TM sequence includes helix-breaking or polar residues, as well as on the hydrophobic length of the potential helices.

## Results

**The glycosylation mapping assay**. For a trapped polypeptide within the ribosome-translocon complex, the number of residues required to bridge the distance between the ribosomal P-site and the active site of the OST can be conveniently measured by glycosylation mapping[9,14]. Radiolabeled, fully assembled translocation intermediates can be prepared in vitro by translating truncated mRNAs (lacking a 3′ stop codon within the coding region) in the presence of [$^{35}$S] amino acids and dog pancreas microsomes. A ribosome halts when it reaches the end of such an mRNA, but the nascent chain remains tethered to the ribosomal P-site because the absence of a stop codon prevents normal termination from occurring (Fig. 1a). A series of neighboring truncation points on the mRNA are tested such that a unique Asn-Ser-Thr (NST) acceptor site for N-linked glycosylation is moved from a position 63 residues to a position 73 residues away from the P-site. The degree of glycosylation is measured for each translation product. N-glycosylation of a nascent chain is detected by an increase in molecular mass of about 2.5 kDa relative to the observed molecular mass of the non-modified molecule.

**TM but not soluble helices have a compact conformation**. We have previously shown that a minimum distance of ~64 residues from the C-terminus of a tethered nascent chain is required to bridge the P-site and the OST active site for sequences with extended conformation (from the extramembranous C-terminal domain of wild-type Lep). This P-NST distance is increased to ~70 residues when model hydrophobic helical stretches are analysed[9,14], suggesting a more compact, likely α-helical conformation. For a fully extended nascent chain (~3.4 Å per residue), more than 12 residues need to be folded to an α-helix (~1.5 Å per residue) to account for this observed change in P-NST distance of ~6 residues. Compaction of the nascent chain positions the glycosylation acceptor site closer to the membrane so that the acceptor asparagine is no longer accessible to the OST active site (see Fig. 1a). This arrangement is in line with the recently reported structure of mammalian ribosome-Sec61-OST complexes[21].

In the current study, glycosylation mapping experiments were performed for nascent chains containing native helical sequences from the VSV-G protein or gp41 TM segments (Supplementary Fig. 1). Glycosylation profiles were obtained that suggested compacted conformations (Fig. 1b, upper panels). Nascent chains harboring non-TM (soluble) helices of comparable lengths (Supplementary Fig. 1, 2a, b), either from an exceptionally stable helix from ribosomal protein L9[22] or from a highly hydrophilic N-acetylglutamate kinase (NAGK)[23] (Fig. 1b, bottom panels), however, displayed a glycosylation pattern suggestive of an extended conformation for these sequences. These striking differences between the glycosylation patterns for the two types of helical sequences (Fig. 1c) indicate that TM helices may fold inside the ribosome exit tunnel, while the nascent polypeptide chains harboring soluble helical sequences remain in an extended conformation. It should be noted that the folding event occurs far inside the ribosome exit tunnel (proximal to P-site), as the putative helix-forming sequences present in our constructs for P-

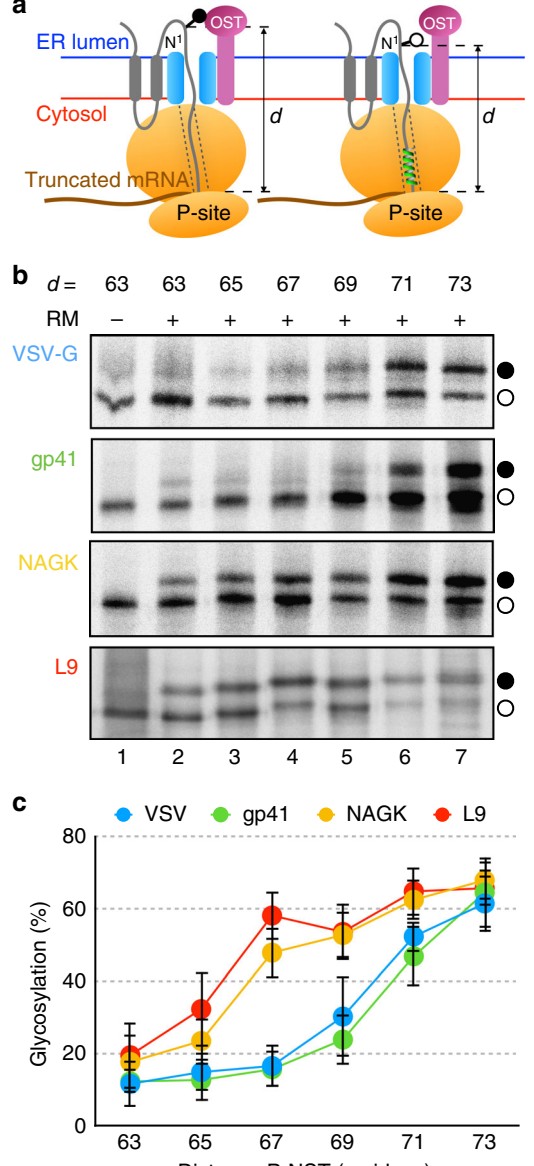

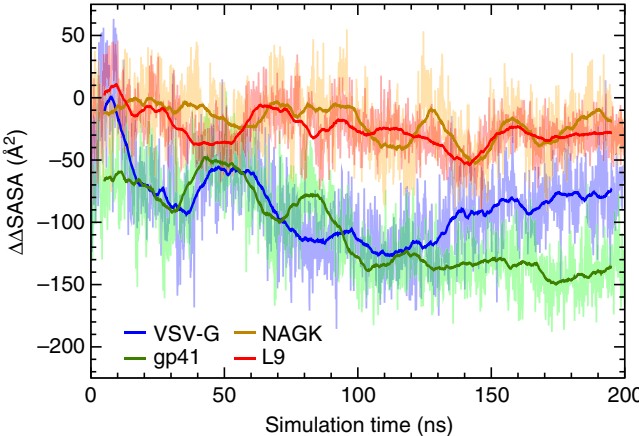

**Fig. 2** Solvent accessible surface area (SASA) for folded versus extended states. Effect of ribosome on solvent accessible surface area (SASA) of hydrophobic residues within the α-helical sequences for folded versus extended states. $\Delta\Delta$SASA = $\Delta$SASA$_{folded}$ − $\Delta$SASA$_{extended}$ (see Methods for the definition of $\Delta$SASA). Negative $\Delta\Delta$SASA values indicate that the ribosome is stabilizing hydrophobic regions of the helical sequence in the folded, α-helical (compact) state more than in the extended state. Conversely, positive values indicate that the ribosome is stabilizing hydrophobic regions of the helical sequence in the extended state more than in the α-helical state

**Fig. 1** Helices in the ribosome exit tunnel. **a** The model protein used in this study (*E. coli* Lep) has two TM segments (gray) and a large C-terminal domain. Ribosome-bound truncated nascent chains of different lengths are generated by in vitro translation, in the presence of dog pancreas microsomes, of mRNAs lacking a stop codon (brown). The minimum number of residues required to span the distance between the ribosomal P-site and the active site of the OST (*d*, distance P-NST) will depend on the compactness of the polypeptide region located inside the ribosome tunnel. Ribosome cartoon is not drawn to scale with respect to the length of the nascent polypeptide chain nor to the membrane thickness. **b** In vitro translation in the absence (−) and presence (+) of rough dog pancreas microsomes (RM) of truncated mRNAs of different lengths harboring the sequences encoding different helices: VSV-G TM segment (residues 463–482), gp41 TM segment (residues 684–705), NAGK helix (residues 5–26), and L9 helix (residues 45–67). The number of residues between the Asn residue in an Asn-Ser-Thr glycosylation acceptor site and the C-terminal end of the nascent chain are shown on top. Glycosylated and non-glycosylated molecules are indicated by black and white dots, respectively. **c** Glycosylation profiles for constructs of the indicated lengths harboring the different helical sequences. Error bars represent the mean ± SD; n ≥ 3. Source data are provided as a Source Data file

NST distances of 67 residues are located at 7–9 residues from the C-terminus of the peptidyl-tRNA (Supplementary Fig. 1). To demonstrate that the ribosomally non-compacting helical sequences from NAGK and L9 are soluble and not capable of inserting into the microsomal membranes in our experimental system, these constructs were analysed using a well-established assay for quantifying the efficiency of membrane integration of tested sequences[24,25]. As expected, translation products of both these constructs revealed no membrane insertion (Supplementary Fig. 3).

Next, we carried out 200-ns simulations of the 67-residue nascent peptide sequences VSV-G, gp41, NAGK, and L9 inside the mammalian ribosome exit tunnel starting from both helical and extended states. Although the conformational space of the peptide is unlikely to have been fully sampled in 200 ns, the initial position of the nascent polypeptide was based on that already present in the cryo-EM structure[26] (Methods), which is similar to that of a nascent peptide in a translating ribosome[27]. For all systems, we measured $\Delta$SASA, the reduction in the solvent accessible surface area (SASA) of the hydrophobic residues within the region of the nascent polypeptide known to be α-helical in its final folded form due to contacts with the hydrophobic residues in the ribosome exit tunnel (see Methods). Thus, $\Delta$SASA represents the degree to which hydrophobic contacts within the tunnel stabilize the nascent polypeptide compared to water; these contacts lower its free energy by ~0.015 kcal/mol Å$^2$ [28]. A more negative $\Delta$SASA indicates that the nascent polypeptide is more stable within the exit tunnel.

The values of $\Delta$SASA for both helical and extended states for all four nascent polypeptides inside the ribosome exit tunnel are plotted in Supplementary Fig. 4. To compare the relative stability of the helical state for the four nascent peptide sequences, we calculated the difference between $\Delta$SASA in the helical and extended states: $\Delta\Delta$SASA = $\Delta$SASA$_{helical}$ − $\Delta$SASA$_{extended}$. After 100 ns, $\Delta\Delta$SASA for the TM and soluble sequences begin to separate. For the TM

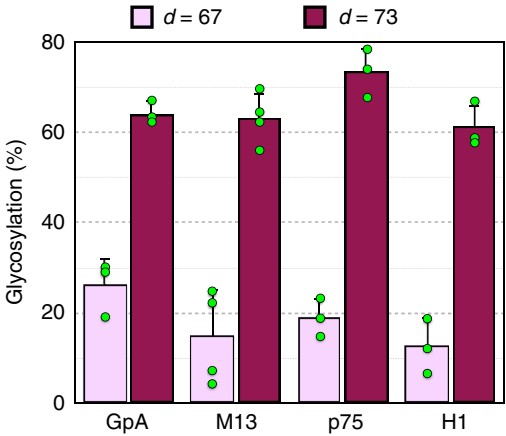

**Fig. 3** TM helices from different origins display a compact conformation. Glycosylation percentage of nascent polypetides with 67 (pink) or 73 (purple) residues between the acceptor Asn and the polypeptide C-terminus. Error bars represent the mean ± SD; $n \geq 3$. Individual data points are shown as green dots. Sequences and full glycosylation patterns can be found as Supplementary Fig. 7. Source data are provided as a Source Data file

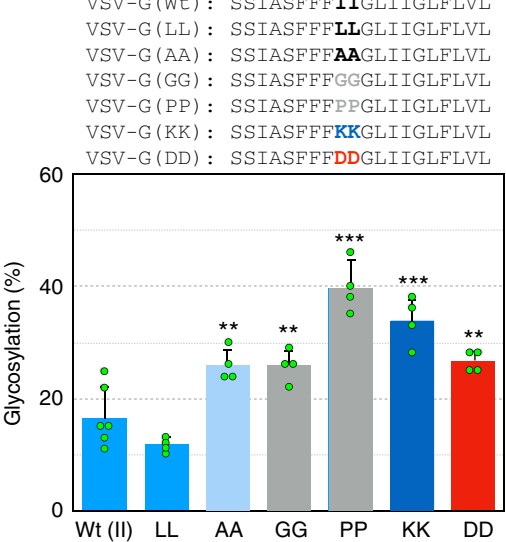

**Fig. 4** Hydrophobicity and helicity affect TM folding. In vitro translation of truncated VSV-G constructs (distance P-NST of 67) in which the central Ile pair (shown in bold) was mutated to less hydrophobic, charged (basic and acid residues shown in dark blue and red, respectively) and helix breaking residues (shown in gray). The average glycosylation percentage is plotted for each mutant. Error bars show the standard deviation of four or more independent experiments (p-values for the comparison with wild type (Ile pair): **<0.01 and ***<0.001). Individual data points are shown as green dots. Source data are provided as a Source Data file

sequences gp41 and VSV-G, ΔΔSASA is negative, indicating that the helical state is stabilized by hydrophobic contacts in the ribosome exit tunnel more than the extended state (see Fig. 2). For the soluble sequences NAGK and L9, ΔΔSASA is close to zero or only slightly negative, indicating that the extended state is neither favored nor disfavored compared to the folded state. The last 100 ns of each system is visualized in Supplementary Fig. 5. For the TM sequences, given the high number of possible hydrophobic residues in the helical region, there are a larger number of possible hydrophobic contacts with the ribosomal proteins L4 and L17 at the constriction point of the tunnel, as well as a smaller number of possible contacts with L39 near the exit pore. In the extended state, however, some of these residues extend out into the exit pore, exposing them to water (see Supplementary Fig. 5a). Pande and coworkers found that large hydrophobic residues such as isoleucine and tryptophan experience a large free energy barrier as they exit the tunnel of an archaeal ribosome[29]. Extended conformations of the TM helices could be destabilized due to a free energy penalty of exposing large, hydrophobic side chains to the solvent. This free energy penalty would be lower for the soluble sequences as they contain fewer large, hydrophobic residues that extend out into the exit pore. Hydrophobic contacts within the tunnel as well as a large free energy barrier upon exiting the tunnel could explain why TM helices are stabilized within the ribosome. Similar results were found for a bacterial ribosome (see Supplementary Fig. 6), suggesting that these contacts are conserved across all domains of life.

To test whether the folding into helices in the ribosome is a common feature for TM segments, we extended our studies to include glycophorin A (GpA), the first hydrophobic domain from Lep (H1), p75 neurotrophin receptor (p75) and the small coat protein of M13 phage (M13) TM sequences. As the maximal differential effect on glycosylation efficiency typically observed for the truncated Lep constructs is at a P-NST distance of 67 residues, we define this distance as the critical number of residues required to distinguish between extended and compact conformations. This was compared with a P-NST distance of 73, which is long enough to be fully glycosylated in all the constructs[9]. As shown in

Fig. 3, all four constructs harboring TM sequences show a clear difference in the glycosylation efficiency for both distances, consistent with the adoption of α-helical structure in the ribosome tunnel of these test sequences. In fact, the observed glycosylation patterns (Supplementary Fig. 7) mirror those obtained for the VSV-G and gp41 containing constructs (Fig. 1b, top panels).

**Determinants of TM helix folding inside the ribosome**. To further study the determinants of TM folding, the mid-region of the VSV-G TM sequence was mutated to code for amino acids designed to affect either hydrophobicity or helicity. When the central Ile pair was conservatively replaced with a Leu pair no significant difference in glycosylation efficiency at a P-NST distance of 67 residues was observed (Fig. 4). However, reducing the hydrophobicity by mutating to an Ala pair resulted in a modest but significant increase in glycosylation efficiency (from 16 ± 5% for the wild type to 26 ± 3%, ± denotes the standard deviation of four or more independent experiments) indicating a less compact conformation. The increase in glycosylation efficiency was even more pronounced when charged residues were engineered into the sequence. Hence, constructs containing either Lys or Asp pairs displayed a glycosylation level (34 ± 4% and 27 ± 2%, respectively) indicative of a more extended conformation (Fig. 4). Similarly, when the central Ile pair was replaced with helix breaking residues, either a Gly pair or a Pro pair, an increased level of glycosylation was observed. The Pro pair led to the greatest increase in glycosylation efficiency (40 ± 5%). From these results, it can be concluded that TM sequence folding inside the ribosome exit tunnel depends on both hydrophobicity and helicity.

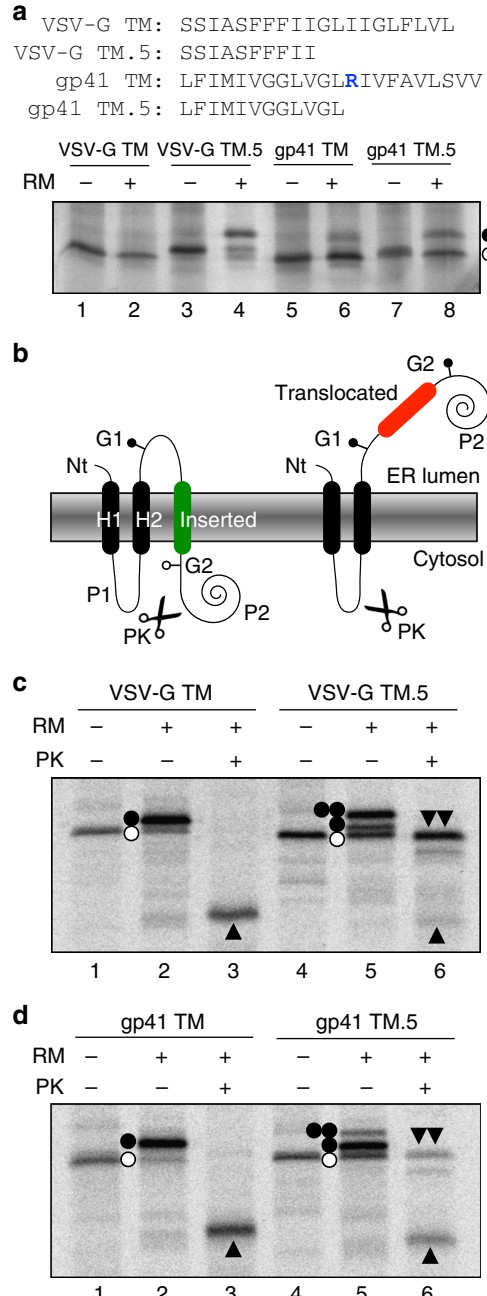

**Fig. 5** Folding depends on hydrophobic length and correlates with insertion. **a** In vitro translation in the absence (−) and presence (+) of dog pancreas rough microsomes (RM) of truncated mRNAs of the same length (distance P-NST 67) harboring VSV-G full length (lanes 1 and 2) or half (VSV-G TM.5, lanes 3 and 4) TM segment, or gp41 full length (lanes 5 and 6) or half (gp41 TM.5, lanes 7 and 8) TM segment. Glycosylated and non-glycosylated molecules are indicated by black and white dots, respectively. Amino acid sequences included are shown on top. **b** Schematic of the engineered leader peptidase (Lep) model protein. Lep, consisting of 2 TM segments (H1 and H2) and a large luminal domain (P2), inserts into RMs in an N$_{lum}$-C$_{lum}$ orientation. In vitro protein translation in the presence (+) or absence (−) of rough microsomes (RM) and proteinase K (PK) of VSV-G (**c**) or gp41 (**d**) derived sequences. Non-glycosylated protein bands are indicated by a white dot; single and double glycosylated protein bands are indicated by one or two black dots, respectively. An upwards black triangle indicates small protected singly glycosylated H2/inserted fragment. A double downward black triangle indicated large doubly glycosylated H2/G1/trasnlocated/G2/P2 fragment. Source data are provided as a Source Data file

**TM stretches that do not integrate do not fold**. If TM sequences are responsible for the folding in the ribosome detected by gly-cosylation mapping, it is possible that the length of the hydrophobic sequence will influence folding events. The contribution to the overall length of a canonical α-helix per amino acid is 1.5 Å. Therefore, a stretch of ~20 consecutive hydrophobic amino acid residues are required to span the 30 Å of the hydrocarbon core of a 'typical' biological membrane[30]. Indeed, the most prevalent length for TM helices is 21 amino acids, according to structure-based statistical analysis[31]. To investigate the relevance of the integrity of the TM sequences in terms of length, constructs were designed containing roughly half of the TM sequence of VSV-G or gp41. As shown in Fig. 5a, when truncated nascent chains of the same length (P-NST distance of 67 residues) containing only half of VSV-G TM sequence (VSV-G TM.5) were translated in the presence of microsomal membranes efficient glycosylation was evident (lane 4). A similar observation was made for the construct harboring roughly half the hydrophobic sequence of the gp41 TM segment (Fig. 5a, compare lanes 6 and 8). These data suggest that short hydrophobic sequences adopt a more extended conformation. Hence, the ribosome exit tunnel can apparently distinguish between legitimate TM segments and a shorter stretch (~10) of non-polar residues in a nascent poly-peptide. It should be noted, however, that the position of the short hydrophobic sequences in the ribosomal tunnel will likely be further from the P-site than in the original TM sequences due to the potentially extended conformation of the C-terminal residues.

Finally, to investigate any correlation between TM helix folding and insertion we measured the efficiency of membrane integration of these sequences into microsomal membranes. Briefly, the host model protein (Lep) consists of two TM segments (H1 and H2) linked by a cytoplasmic region (P1) and a large C-terminal domain (P2), and inserts into ER-derived RMs with both ends translocated across the membrane and then, oriented toward the lumen (Fig. 5b). The TM-derived sequences were engineered into the large P2 domain and flanked by two acceptor N-linked glycosylation sites (G1 and G2). The engineered glycosylation sites can be used as membrane insertion reporters because, in the full-length protein, G1 will always be glycosylated due to its native luminal localization, but G2 will only be glycosylated upon translocation of the test region into the lumen of the microsomes. A singly glycosylated construct in which the test sequence is inserted into the membrane has an increased apparent molecular mass of ~2.5 kDa relative to the observed molecular mass of the constructs expressed in the absence of microsomal membranes; double glycosylation (i.e., membrane translocation of the sequence analysed) increases the apparent molecular mass by ~5 kDa. When proteinase K (PK) is added to microsomes, it digests the non-glycosylated form of the P2 domain that is exposed to the cytoplasm (Fig. 5b, left) generating a small protected fragment (H2/G1/inserted); alternatively, it produces a large protected (H2/G1/translocated/G2/P2) fragment when the P2 domain is located in the lumen of the microsomes (Fig. 5b, right). In the presence of microsomal vesicles, the translation of constructs harboring the complete VSV-G TM segment resulted in singly glycosylated forms of the protein (Fig. 5c, lane 2). PK treatment of these samples yielded a small protected singly glycosylated H2/G1/inserted fragment (lane 3), indicating that the full length VSV-G TM sequence was properly inserted into the membrane. Comparable results were obtained with constructs harboring the gp41 complete TM sequence (Fig. 5d, lanes 1–3). However, the translation of constructs containing only half of the VSV-G hydrophobic sequence (VSV-G TM.5) in the presence of microsomes resulted mainly in double glycosylated (translocated) forms (69%, Fig. 5c, lane 5). Furthermore, following PK

treatment, which degrades membrane protein domains located exclusively towards the cytosol, whereas membrane embedded or lumenally exposed domains are protected, a large protected (H2/ G1/translocated/G2/P2) fragment was generated. With the gp41 sequence, it was found that 12 hydrophobic residues are sufficient for integration (81%, Fig. 5c, lane 5) into the membrane (Fig. 5d, lanes 4–6), albeit with a reduced efficiency compared to the complete TM sequence (Fig. 5d, lanes 1–3). Interestingly, although the N-terminal half of the gp41 TM sequence displayed a more helical structure compared to the same region of VSV-G in our MD simulations (Supplementary Fig. 8), we note that both folded TM sequences remain in compact, near-helical states (Supplementary Fig. 9). All these data are consistent with the less compact conformation suggested in the glycosylation distance mapping experiments (Fig. 5a), and with previous data on the insertion of short hydrophobic sequences[25,32]. Thus, the folding of TM helices appears to precede engagement with the translocon and subsequent insertion into the membrane.

## Discussion

The ribosome exit tunnel (in both prokaryotes and eukaryotes) is ~100 Å in length, varying in diameter between a maximum of 20 Å distal the P-site and a minimum of 10 Å at its narrowest constriction, ~30 Å from the P-site[7,8]. There are a number of previous studies demonstrating that the environment of the ribosomal tunnel is permissive for the adoption of α–helical secondary structure of some nascent polypeptide chains[9,11,15,16,33,34] along its entire length, even at the constriction[35] and very close to the ribosomal P-site[36]. It has also been shown that limited tertiary folding can take place, but possibly only in the wide exit domain region of the tunnel distal to the P-site[37–40]. However, not all sequences that adopt α-helical structure in the folded protein necessarily form within the ribosome.

In this study, we have investigated the intrinsic ability of test sequences in the context of stalled nascent polypeptide chains to attain a compact structure within the ribosome. Test peptide sequences with known stable helical propensity in their native folds from both integral membrane proteins (hydrophobic helices) and soluble proteins (hydrophilic helices) were used. A principal finding was that when hydrophobic test sequences were present within the ribosomal tunnel there was clear evidence of compaction, as demonstrated by reduced glycosylation efficiency at the 'critical' distance, whereas there was no such indication for hydrophilic test sequences despite their known α–helical propensity. All of the six hydrophobic test sequences, namely TM domains of integral membrane proteins (VSV-G, gp41, GpA, LepH1, p75, and M13), showed compaction, indicating that nascent chain α–helix accommodation in the ribosome is a common, if not general phenomenon. This is in agreement with earlier studies using FRET to measure compaction of hydrophobic and hydrophilic sequences showing that hydrophobic sequences compacted whereas hydrophilic sequences did not[15]. However, the hydrophilic sequence used in this earlier study was not specifically selected on the basis of helical propensity, raising the possibility that lack of compaction was simply a property of a non-helical sequence being tested. The positioning of the test sequences in the FRET study was closer to the P-site (four residues away) than in our current study in which the start of the test sequences is located 7–9 residues from the P-site (for nascent chains with $d = 67$ residues). This suggests that adoption of α–helical conformation for hydrophobic helices is a very early event initiated 20–30 Å away from the P-site. Further supportive evidence for early helical adoption of non-polar helices comes from elegant studies on polyalanine and Kv1.3 TM6 channel helix as test sequences and a PEGylation method to probe for

helicity[16,33]. Other studies, however, reveal compaction only at a site distal from the P-site in the ribosome exit vestibule[10].

What is it about the environment within the ribosome that induces hydrophobic, but not hydrophilic, helices to adopt helical conformation? It has been previously hypothesized that non-polar areas within the tunnel may provide a surface for hydrophobic helices to nucleate upon[41]. The cryo-EM structure does not reveal any obvious extensive hydrophobic surfaces for this to happen[42]. To try to address the above question molecular dynamics simulations of the test peptides in the ribosome were performed. The main difference observed between test sequences was a reduction in solvent accessible surface area of hydrophobic test sequences in comparison to hydrophilic test sequences when modeled as helices. We propose that the test sequences can all sample conformational space but that hydrophobic helices are stabilized by subtle non-specific hydrophobic side chain interactions within the ribosome tunnel whereas hydrophilic helices are not. The simulations suggest that hydrophobic amino acids within ribosomal proteins provide hydrophobic contacts with amino acid residues in hydrophobic helices. Alternatively, it has been shown theoretically that space confinement within a cylinder resembling the ribosomal tunnel can entropically stabilize α-helices without invoking-specific interactions with the tunnel wall[43].

Hydrophobic stabilization of TM helices was observed for both a eukaryotic and bacterial ribosome model. It has to be taken into consideration that, as nascent chains are elongating, the tunnel environment of stretches of amino acids will change. Although evidence for compaction of the hydrophobic test sequences within the ribosome at particular positions is strong, the experimental system analyses stalled complexes and does not provide information on conformations at other tunnel locations. However, together with other data suggesting that hydrophobic helices can form in the vicinity of the P-site[35,36], and that helices are permissible along the whole length of the tunnel,[44] we hypothesize that hydrophobic helices could maintain their helicity throughout translation into the translocon and integration into the membrane. This, however, remains to be tested experimentally for the test sequences used in this present study.

In the experiments where two Ile residues in the mid region of VSV-G were replaced with amino acid residues intended to disrupt helicity or reduce overall hydrophobicity there was in general a loss of compaction. There were, however, quantitative differences with Pro-Pro > Lys-Lys > Gly-Gly > Asp-Asp > Ala-Ala with respect to increasing glycosylation efficiency, a measure of loss of helicity. Furthermore, an incomplete VSV-G TM domain did not compact, in good agreement with previous FRET measurements[15]. Although, we do not know the precise number of residues in helical conformation, we estimate that at least 14 hydrophobic residues from the VSV-G TM sequence appear to be needed to display a glycosylation pattern compatible with α-helix formation, as well as for membrane integration (see Supplementary Fig. 10). These results suggest that a long hydrophobic stretch of amino acids is a requirement to facilitate stable helix formation in the ribosome. This does not rule out localized transient short helix formation but indicates that a long helix can make more substantial contacts and/or increase entropic stabilization[43], sufficient to stabilize the helical conformation. There must be some cooperativity of interactions within the helix, or with the walls of the ribosome tunnel that facilitate the stabilization. In our assay, if there is conformational flexibility with switching between helical and non-helical structures we would expect glycosylation to occur during periods when the peptide is non-helical. Hence, only highly stable compaction is detected in our experimental system.

Is there a physiological relevance for hydrophobic helices forming within the ribosomal tunnel? For spontaneous insertion of integral membrane proteins into membranes it has been proposed that proteins must adopt helical conformation pre-insertion in order to overcome unfavorable energetic barrier of exposing polar peptide bonds to the hydrophobic interior of a lipid bilayer[17,45]. It is possible that this is also the case for translocon-assisted insertion and that folding in the ribosome enhances recognition of TM helices for integration. Without the early adoption of a helical conformation, TM domains would enter the transolocon with exposed peptide bonds which may negate recognition for integration. Whereas the proteins can exist in a helical conformation in the translocon tunnel[7,14] it is not clear whether the environment of the translocon would compel helical folding.

In summary, we conclude that overall hydrophobicity, helicity and length are major determinants of α–helical adoption within the ribosomal tunnel. This could facilitate recognition by SRP[46] and/or a favorable conformation for membrane integration upon entering the translocon.

## Methods

**Enzymes and chemicals.** All enzymes, as well as plasmid pGEM1, TNT T7 Quick Coupled System and rabbit reticulocyte lysate were from Promega (Madison, WI, USA). ER rough microsomes from dog pancreas were from tRNA Probes (College Station, TX, USA). EasyTag™ EXPRESS$^{35}$S Protein Labeling Mix, [$^{35}$S]-L-methionine and $^{35}$S-L-cysteine, for in vitro labeling was purchased from Perkin Elmer (Waltham, MA, USA). Restriction enzymes and Endoglycosidase H were from Roche Molecular Biochemicals (Basel, Switerland). Proteinase K was from Sigma-Aldrich (St Louis, MO). The DNA plasmid, RNA clean-up and PCR purification kits were from Thermo Fisher Scientific (Ulm, Germany). All oligonucleotides were purchased from Macrogen (Seoul, South Korea).

**DNA manipulation.** The helical sequence from NAGK (residues 5–26, Uniprot #Q9HTN2) and L9 (residues 46–67, Uniprot #P02417) and the TM sequence from VSV-G (residues 463–482, Uniprot #P04884), gp41 (residues 684–705, Uniprot #P03375), GpA (residues 92–114, Uniprot #P02724), M13 (residues 21–43, Uniprot #P03618), Lep H1 (residues 1–22, Uniprot #P00803) and p75 (residues 251–272, Uniprot #P08138) were introduced into the modified Lep sequence into the plasmid pGEM-1 between the *Bcl*I and *Nde*I sites by PCR-amplification sequence with primers containing appropriate restriction sites (see Supplementary Table 1). After PCR amplification, the PCR products were purified with Qiagen Purification Kit (Hilden, Germany). Purified PCR and Lep vector were digested simultaneously with *Bcl*II/*Nde*I for 3 h at 37 °C and then purified on agarose gel 1%, followed by a purification with Qiagen Band Extraction (Hilden, Germany). PCR purified products and vector digested were ligated overnight by T7 DNA ligase (New England Lab, USA), the constructs were electroporated into DH5α cells. Positive clones were selected on ampilicilin plates (100 μg/ml) and verified by sequencing (Macrogen Company, South Korea)[47].

Full-length *Lep* DNA was amplified directly from the pGEM1 plasmid using a reverse primer with a stop codon at the end of the *Lep* sequence: 5′-CTATTAatggatgccgcc-3′[48,49]. Alternatively, we prepared templates for the in vitro transcription of each truncated mRNA (without stop codon) at the 3′-end[9,14]. Truncated constructs were obtained by using forward primer that include the T7 promoter sequence at the 5′-end: 5′-atagtaTAATACGACTCACTATAGGGaaaccaccatggcgaatatg-3′. The 3′ reverse primers (see Supplementary Table 1) were designed to have approximately the same annealing temperature as the 5′ forward primer, without stop codons and annealed at specific positions to obtain the stalled polypeptide nascent chains containing the appropriate P-NST distances (*d*)[9,14,50].

Agarose gels (2%) were used to verify PCR product size then samples were cleaned using the Wizard® SV Gel and PCR Clean-up System (Promega, Madison, WI) following the manufacturer's recommendations.

**In vitro transcription and translation.** The Lep-derived constructs and truncated constructs were transcribed and translated using the TNT T7 Quick Coupled System (Promega, Madison, WI). The reactions contained 75 ng of DNA template, 0.5 μl of EasyTag (5.5 μCi), and 0.25 μl of microsomes (tRNA Probes, College Station, TX) were incubated for 40 min at 30 °C. The translation products were ultracentrifuged (100,000×*g* for 15 min) on a sucrose cushion, and analysed by SDS-PAGE. The bands were quantified using a Fuji FLA-3000 phosphoimager and Image Reader 8.1j software.

For the proteinase K protection assay, the translation mixture was supplemented with 1 μL of 50 mM CaCl$_2$ and 1 μl of proteinase K (2 mg/ml), and the digestion reaction was incubated for 40 min on ice[47,51]. Before SDS-PAGE

analysis, the reaction was stopped by adding 1 mM phenylmethanesulfonyl fluoride (PMSF). All the translation/glycosylation experiments were repeated at least four times.

**Molecular dynamics simulations.** The structure of an actively translating human ribosome and its nascent polypeptide chain were taken from PDB 5AJ0[26]. A reduced model was created in a manner identical to that used for the bacterial ribosome previously[27]. All residues of the ribosome further than 25 Å from the nascent polypeptide were excluded, whereas residues between 23 and 25 Å away from the tunnel were held in place using harmonic restraints for all simulations. Given the high degree of solvation and ion content of the ribosome, the Bjerrum length at which long-range interactions start to become negligible is expected to be close to that of water (~7 Å), much shorter than the radius of exclusion (25 Å). Non-helical regions of the peptide sequences were fit to the nascent peptide sequence, while helical regions were inserted as ideal α-helices. For extended systems, the N-terminal residue was held in place with a harmonic restraint of 5 kcal/mol Å$^2$. The visualization and analysis program VMD[52] was used to solvate the system with 31,058 TIP3P[53] water molecules in a 99 × 91 × 151 Å$^3$ box. The system was then neutralized with 0.1 M KCl, resulting in a final system size of ~120,000 atoms.

MD simulations were carried out using NAMD 2.12[54] running on both CPUs and GPUs. The CHARMM36 force field was used for proteins[55] and nucleic acids[56], including modified bases[57]. The temperature was maintained at 300 K using Langevin dynamics; the pressure was kept at 1 atm using the Langevin piston method[58]. The equations of motion were integrated using the RESPA multiple time-step algorithm with a time step of 2 fs for all bonded interactions, 2 fs for short-range non-bonded interactions, and 4 fs for long-range electrostatic interactions. Long-range electrostatic interactions were calculated using the particle-mesh Ewald method[59]. Bonds involving hydrogen atoms were constrained to their equilibrium length. Initially, all non-solvent atoms were restrained as the solvent was free to relax for 1 ns. Side-chain atoms were then released and equilibrated for another 1 ns. Backbone restraints were slowly dropped to zero for another 1–5 ns until the system was stable. Finally, additional restraints were added to maintain the helical segment of the nascent polypeptide in an ideal α-helix for 10 ns. Production equilibrium simulations were performed with only the outer 23–25 Å of the ribosome restrained.

ΔSASA was determined by first calculating the solvent accessible surface area (SASA) of the hydrophobic residues of the α-helical region of the nascent polypeptide inside the ribosome exit tunnel and then subtracting the SASA of the hydrophobic residues of the same region alone. This difference reveals how much of the hydrophobic surface area of the nascent polypeptide is concealed by hydrophobic contacts with the ribosome.

## Data availability

Data supporting the findings of this manuscript are available from the corresponding author upon reasonable request. A reporting summary for this article is available as a Supplementary Information file. The source data underlying Figs. 1b, c, 3, 4, 5 and Supplementary Figs 3b, 7 and 10 are provided as a Source Data file.

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

## Acknowledgements

We thank members of the Mingarro lab for discussions. This work was supported by grants BFU2016-79487-P from the Spanish Ministry of Economy and Competitiveness (MINECO, ERDF supported) and PROMETEO/2014/061 from the Generalitat Valenciana (to I.M.), and by grant MCB-1452464 from the U.S. National Science Foundation (to J.C.G.). M.B.-P. was recipient of a postdoctoral fellowship from the Generalitat Valenciana. C.B.-D. was recipient of a predoctoral fellowship from the MINECO (FPI program). S.T. and B.G. were recipient of a predoctoral fellowship from the University of Valencia (V Segles program and Atracció de Talent program, respectively). Computational resources were provided via the Extreme Science and Engineering Discovery Environment (XSEDE), which is supported by NSF grant number OCI-1053575.

## Author contributions

M.B.-P., C.B.-D., S.T., and B.G. performed the translation/glycosylation experiments. A. H. and J.C.G. carried out the MD simulations. M.B.-P., C.B.-D., S.T., A.H. B.G., I.N., P. W., J.C.G., G.v.H., and I.M. analyzed the data. I.M. conceived and planned the investigation. P.W., J.C.G. and I.M. prepared the manuscript with discussions and improvements from all authors.

## Additional information

**Competing interests:** The authors declare no competing interests.

