## [Peer Review File · Nature Communications]

Reviewers' Comments:

Reviewer #1:

Remarks to the Author:

The manuscript addresses the influence of the ribosomal peptide exit tunnel on the conformation of the nascent peptide inside the tunnel. Previous FRET measurements (ref. 14 in this manuscript) suggested that a transmembrane segment assumes a compact, presumably alpha-helical structure in the exit tunnel near the peptidyl transferase center and moves through the tunnel in this conformation. In contrast, a nascent secretory protein is in an extended conformation in the exit tunnel.

The present manuscript takes the story several steps further. Potential limitations of the previously used FRET approach are avoided by using an elegant glycosylation assay for monitoring changes of the apparent length of nascent peptides within the exit tunnel due to folding. The results confirm that hydrophobic TM segments in the tunnel assume a compact, likely helical conformation within the tunnel and potential helices taken from soluble proteins do not. Furthermore, molecular dynamics simulations suggest that interactions between particular residues of the nascent chain and elements of the tunnel wall stabilize the helical conformation of TM segments. Finally, modulation of the hydrophobicity and helix-forming propensity by introducing mutations into a particular TM segment supports the conclusion that it is alpha helix formation that causes the observed effects.

Taken together the results of the paper are suitable for publication in Nature Communications. However, there are a few points that should be amended or clarified before the manuscript may be accepted. These are detailed in the following.

The translation and glycosylation assays of stalled ribosome-nascent-chain complexes shown in Fig. 1b are quite variable. For example, the efficiency of gp41 translation in the presence of microsomes (lane 2) appears rather low, and the same is true for NAGK (lanes 1 and 2). It is not obvious in these and other cases that the quantification of the gel depicted in Fig. 1c is unambiguous. Given the central role of the assay, it is important to clarify this problem, preferably by providing better data.

The selection of ribosome residues included in the box employed for molecular dynamics simulations should be justified more explicitly. Furthermore, if I understand correctly, coordinates from *E. coli* ribosomes were used for the simulations, although mammalian components were used for the experiments. Why not use coordinates from eukaryotic ribosomes? Despite the high level of conservation, this should be explained.

The simulations show substantial differences in the parameters between alpha-helical and extended states of the nascent peptides, as expected. There are only small changes of these parameters over the simulation time of 100 ns, indicating there are essentially no fluctuations in this time window. Is there information from longer simulations?

Reviewer #2:

Remarks to the Author:

This study aims to characterize determinants of helix formation, which could impact interaction with chaperones and SRP for efficient ER integration of membrane proteins and translocation of non-membrane proteins. The approach uses a glycosylation assay that is now standard to estimate the distance spanned by a test peptide segment (~ 20-22 residues) engineered into a Lep nascent chain in the ribosome-translocon-peptide complex. This distance reflects the conformation (compact or extended) of the test segment. The method has been used by the von Heijne group since its first introduction in 1996-2000 (Mingarro et al., BMC Cell Biology 1:3 (2000); Whitley et

al., JBC 271, 6241(1996)). In the current manuscript, the authors use a series of test sequences, primarily 2 transmembrane (TM) segments (VSV-G and gp41), plus 4 additional TMs (more limited investigation), and 2 non-membrane test peptides. All test sequences are helical in the mature, full-length native protein. In addition to the glycosylation assay, the authors verify insertion (or lack thereof) of the fragments into the ER membrane with a simple PK-digestion assay, and molecular dynamics simulations of solvent accessible surface area (SASA) to indicate that TMs VSV-G and gp41 in the ribosome tunnel display a greater reduction in SASA (Δ SASA) compared to non-TM segments. The results suggest that TM segments, but not non-TM segments, of a given hydrophobicity and length, form helices in the ribosome tunnel. These findings and conclusions add to the repertoire of conformational determinants and governance of membrane insertion. Several published work, including the authors', spanning the last 15 years have presented evidence consistent with the current Ms, but nonetheless, this paper adds more detail and clarifies some tenets of secondary structure formation. General comments, followed by specific comments/criticisms are below.

General:

- 1). Although the authors cite Uniprot ID numbers in the Methods for the test segment sequences, and give sequences for 6 of the 8 TM test sequences in the Figures, the authors should show the sequences for the non-TM L9 and NAGK, and the upstream/downstream Lep sequences flanking the test sequences for peptides 63-73 (presumably extended with 3.4Å/aa). This assists the reader in comparatively evaluating the sequences, interpreting the Results, and informing the reader which parts of each test sequence are situated at upstream tunnel locations.
- 2). Clarification is needed for how each construct was made (especially for the 0.5 constructs), the location of each sequence in the tunnel, and what their neighboring sequences are, as both location and neighboring residues are important for helix formation. Are different restriction sites in the mRNA used to change P-NST distance or are NST mutations engineered to change the P-NST distance? The former will change the location of the test sequence and Lep residues. The latter will only move NST while the test sequence remains identical for all constructs. If the latter is the case, then is there an 8aa Lep-linker (as implicated in two places in the manuscript) between the P-site and the start of the test sequence? A change in test sequence tunnel location can impact folding.
- 3). It is not clear how many, nor which, of the 20-22 residues (GpA has 23) in the test sequences, are helical in the tunnel. Fig. 1a (right) depicts all 20-22 residues as helical. But, need this be the case? (Also, 22 helical residues span only 33Å of the tunnel, not the whole 100 Å-tunnel as drawn (see comment 4 below)). In the P-NST = 73 TM helix, if only 11 test residues are helical, glycosylation will still be maximal and the estimated P-NST distance will be 227Å, which is sufficient to span the P-OST requirement of ~ 205Å (Mingarro et al., 2000; Whitley et al., 1996; Ismail et al 2012). Additionally, as the P-NST length is increased, if this is due to truncation at different mRNA codons (which I do not know; please see comment 2, above), then test sequences would be in different tunnel locations. Regardless, the following issues should be considered: How many test-sequence residues are helical, which ones are helical, where do they reside, and at which tunnel location does helix formation occur?
- 4). Figure 1a, right cartoon should be drawn more accurately, e.g., the helix is shown spanning all 100Å of the tunnel and no downstream Lep residues, which we assume are extended. Given comments 2 and 3 above, perhaps there is a better way to represent the relative distances of the green test sequence, the putative α -helix region, and the Lep fragments in the 100Å-tunnel.
- 5). The calculations and explanations for Supplementary Figure 5, and the accompanying text in the Results and Methods, need to be expanded and explained. It is not clear to the reader how the residue distribution along the tunnel (Δ PTC) was determined, nor what assumptions were made. See further comments/questions below under Supplementary Figure 5.

Specific:

Title: The title may be overstated. What does "Far" mean, especially given the distance issues in General comment 3 (above) and the specific comments below? "Far" depends on how many of the test residues form a helix, where in the tunnel this helix forms, and where non-TM test sequences

eventually compact. Stating "...Far Inside the Ribosome Tunnel..." suggests that a positive control, i.e., soluble helices in the distal regions nearer to the exit port do form a helix. But, it is not clear that this control experiment was done. Perhaps a title like "Intrinsic ability of peptide chains to attain a compact structure within the ribosome" would be more appropriate.

Line 87: "non-TM stretches do not attain full helical conformations inside the ribosome." This may be worded a little too strongly since there is evidence in the literature for non-TM sequences being compact in the ribosome, and---see above comment---depends on location. Or do the authors want to imply that the soluble helical test sequences do not fold anywhere along the 100Å-length?

Line 118: "~3.4Å". An estimate of 3.2Å/aa has been used to analyze pulling forces during translocon-mediated membrane integration by these authors in a previous study (Ismail et al. 2012) and a total distance from PTC to OST (P-NST) was determined to be 205Å.

Lines 133-136: As cited in the above comments, it is not clear which part of the test sequence forms a helix and where in the tunnel this happens, e.g., could it be that in the constriction region the C-terminus of the test TM sequence is extended, but has its N-terminal region compact below the constriction?

Lines 174-177: How do the implied stabilizing effects ("higher affinities") between peptide side-chains and the tunnel at different regions from the PTC, shown in Supplementary Fig. 5, compare with Pande's free-energy profiles at different tunnel locations (Petroni et al., 2008)?

Lines 219-220: Regarding the 0.5 TM glycosylation experiments (Fig. 5), the 0.5 segments are the N-terminal residues of the full-length VSV-G or gp41. Where are these 0.5 segments inserted in the Lep peptide? Are they in the identical peptide locations as they are in the full-length peptide, with the deleted C-terminal test residues replaced by additional 10-11 residues from Lep (made by using a new restriction site in the Lep mRNA) that are downstream of the 0.5 segment? Alternatively, is the mRNA restriction site unchanged but the C-terminal 10-11 residues are mutated to be extended? Regardless, what is the sequence of these replacement residues, can they influence folding, and are the N-terminal 0.5 residues in a different tunnel location due to the extended downstream 10-11 Lep residues? This information and clarification should be included.

Line 290: "our current study in which the start of the test sequences is located ~ 8 residues from the P-site". Please clarify if this is only for the 67 P-NST length peptide or for all full-length test constructs, depending on how the constructs were made (see General comment 2), above).

Line 291: Given the above concern about location and number of residues in the helix (General comment 2), it might be better to avoid stating that adoption of a helical structure is initiated at the P-site.

Line 472: Typographical error: should be Supplementary Fig. 6.

Figure 1: Please see General comments 2-4, above, regarding Fig. 1a. In Fig. 1b, why is there an apparent glycosylation band in lane 1 in the absence of RM (no OST; compare lanes 1 and 2)? Is this a background band and if so, does it vary with P-NST length, or is it constant and does not contribute to the intensity of the glycosylation band?

Figure 5: Please see above comment, Lines 219-220. In addition, could the authors comment on why there is a small protected singly-glycosylated H2/inserted fragment, i.e., upward single black triangle, lane 6 in Fig. 5c and d? Does this suggest some fraction of 0.5 TM inserts?

Supplementary Figure 2: What length peptide is used here for each test peptide? 73? 67? This should be consistent with the % glycosylation in lanes 2, 5, 8. In addition to unglycosylated and singly-glycosylated bands, why is there a top band in lane 2(VSV-G, +RM), whereas lane 1 (VSV-

G, -RM) only shows full-length TM peptide? Does this suggest a doubly-glycosylated species is present in lane 2?

Supplementary Figure 3: This approach is quite good and is the basis for the results shown in Fig. 2. With regard to Supplementary Fig. 3, Δ SASA (bottom panel), if as stated above (General comment 3 and Lines 133-136), only some of the VSV-G and gp41 residues (solid lines) are helical and their location with respect to tunnel interaction sites matters, then I would expect the differential $\Delta\Delta$ SASA curves for VSV-G and gp41 versus NAGK and L9 to be less dramatic. Nonetheless, the simulations point out one extreme endpoint, but how does this relate to the glycosylation results and the above consideration?

Supplementary Figure 5: This Figure, and the accompanying text in the Results and Methods, are not adequately explained or defined. For example, is this the 67-residue peptide? What sequences are used in each panel? How is the "Distance from PTC (Å)" calculated, since this depends on the conformation and location of each residue? For each type of residue evaluated, does this represent an in silico mutagenesis of each residue in each peptide? Or a frequency distribution for each type residue in each native test sequence? How were each of the 4 test peptides analyzed to give the distributions shown in Supplementary Fig. 5? Please define the color-coding. Finally, how do the results shown in Supplementary Fig. 5 relate to the functional glycosylation results and implications for compaction, and directly to the test peptide sequences? The above issues need to be explicitly stated in the Results, Methods, Legend, and perhaps the Discussion.

Reviewer #3:

Remarks to the Author:

As I understand it there are two claims in this paper. (1) There is compaction in TM helices in the ribosome throughout the 100 or so angstrom length but is not the case for soluble hydrophobic sequences. (2) The helix formation depends on the length. They do experiments and MD simulations to reinforce these findings. The findings are interesting but the interpretation does not appear clear to me. Please address the following questions if possible.

General Comments: (1) Is there a critical length for helix formation? In other words, if length is key what should the minimum length be for TM helix? (2) It is unclear to me why the authors argue that hydrophilic sequences do not form helices in the tunnel. As I recall, Beckmann (not just ref. 44) found that many sequences (not necessarily TM only) can form compact helices although the extent of helix formation might vary depending on the location in the tunnel. Are the results here not consistent with these older findings? (3) A mechanism suggested by the authors is that there are favorable intramolecular interaction in the hydrophobic TM sequences. It has been shown theoretically (Ziv PNAS vol 102, p18956) that confinement (hence entropy considerations) could stabilize helices. Why this not hold here? In some papers by Deutsch this notion was invoked. (4) In the above cited paper the authors showed that length of the polypeptide chain plays an important role as surmised in this article. However, the mechanism appears to be different.

I am not clear about the MD simulations. How can the authors assure me that they have sampled the conformations of the confined chain to draw statistically reliable conclusions about the equilibrium between helix and unfolded structures? This pertains to Fig. 2 and others in the SI.

There appears to be an inverse correlation between glycosylation efficiency and compaction versus extension. It would be nice get a quantitative relationship, and in turn with the extent of helix formation. Because I am unclear about the MD results this is important for the readers. Some quantitative connection between the major experiments and helix formation is needed.

We thank the reviewers for their thoughtful and very supportive comments (in blue). Below, we address the specific points raised by the reviewers and elaborate on the corresponding changes in the manuscript.

Reviewer #1

The translation and glycosylation assays of stalled ribosome-nascent-chain complexes shown in Fig. 1b are quite variable. For example, the efficiency of gp41 translation in the presence of microsomes (lane 2) appears rather low, and the same is true for NAGK (lanes 1 and 2). It is not obvious in these and other cases that the quantification of the gel depicted in Fig. 1c is unambiguous. Given the central role of the assay, it is important to clarify this problem, preferably by providing better data.

We have provided better gels to clarify the data.

The selection of ribosome residues included in the box employed for molecular dynamics simulations should be justified more explicitly. Furthermore, if I understand correctly, coordinates from *E. coli* ribosomes were used for the simulations, although mammalian components were used for the experiments. Why not use coordinates from eukaryotic ribosomes? Despite the high level of conservation, this should be explained.

We agree with the reviewer that the choice of model system is not ideal. To address this concern, we repeated our simulations for a human ribosome structure for which a nascent peptide chain was present¹ (PDB: 5AJ0); note that a dog ribosome structure is not presently available. We found similar results for our SASA analysis for simulations using the human ribosome (extended to 200 ns) and Fig. 2 has been updated. Results for the bacterial ribosome have been moved to the Supplementary Material (Fig. S6).

We have also added additional justification to the Methods (Page 18, lines 416-419) for our choice of a 25-Å cutoff for the ribosome based on the relatively short Bjerrum length (7 Å) in water, which is the length at which electrostatic interactions start to become negligible.

The simulations show substantial differences in the parameters between alpha-helical and extended states of the nascent peptides, as expected. There are only small changes of these parameters over the simulation time of 100 ns, indicating there are essentially no fluctuations in this time window. Is there information from longer simulations?

We extended all new simulations of the human ribosome structure to 200 ns for the folded and extended states for all four sequences studied, resulting in 1600 ns of total simulation time (in addition to the 800 ns run for the bacterial system). After some initial fluctuations, with the exception of the VSV-G sequence, Δ SASA values are well converged after 100 ns. $\Delta\Delta$ SASA values for

TM VSV-G sequence trend upward during the second half of the simulations, but they are still below the values for the two soluble sequences. Furthermore, gp41 remains significantly lower.

Reviewer #2

General comments

1). Although the authors cite Uniprot ID numbers in the Methods for the test segment sequences, and give sequences for 6 of the 8 TM test sequences in the Figures, the authors should show the sequences for the non-TM L9 and NAGK, and the upstream/downstream Lep sequences flanking the test sequences for peptides 63-73 (presumably extended with 3.4Å/aa). This assists the reader in comparatively evaluating the sequences, interpreting the Results, and informing the reader which parts of each test sequence are situated at upstream tunnel locations.

We have included the sequences for all the constructs (including the upstream/downstream Lep flanking sequences) analysed as new Supplementary Figure 1. To emphasize the non-helical conformation of the residues covering the 63-73 region in our constructs (RLSERKETLGDVT), we show Lep P2 domain structure as new panel (c) in Supplementary Fig. 2, highlighting in pink this region.

2). Clarification is needed for how each construct was made (especially for the 0.5 constructs), the location of each sequence in the tunnel, and what their neighboring sequences are, as both location and neighboring residues are important for helix formation. Are different restriction sites in the mRNA used to change P-NST distance or are NST mutations engineered to change the P-NST distance? The former will change the location of the test sequence and Lep residues. The latter will only move NST while the test sequence remains identical for all constructs. If the latter is the case, then is there an 8aa Lep-linker (as implicated in two places in the manuscript) between the P-site and the start of the test sequence? A change in test sequence tunnel location can impact folding.

The sequences for both 0.5 (TM.5) constructs are now included in new Supplementary Figure 1. As the referee points out, changes in test sequence tunnel location can impact folding. To avoid these changes, for the TM.5 constructs the N-terminal end of the TM helices was kept fixed at the same position (see also our response to Reviewer #3 General Comment 1). Then, the C-terminal half of each TM helix was replaced by non-helical residues from Lep P2 domain (see Supplementary Fig. 2c). We truncated the DNA templates to remove stop codon, thus trapping the translating ribosomes at the last codon, as explained on page 5 (The glycosylation mapping assay). Then, all truncations were performed at appropriate (different) mRNA codons.

3). It is not clear how many, nor which, of the 20-22 residues (GpA has 23) in the test sequences, are helical in the tunnel. Fig. 1a (right) depicts all 20-22 residues as helical. But, need this be the case? (Also, 22 helical residues span only 33Å of the tunnel, not the whole 100 Å-tunnel as drawn (see comment 4 below)). In the P-NST = 73 TM helix, if only 11 test residues are helical, glycosylation will still be maximal and the estimated P-NST distance will be 227Å, which is sufficient to span the P-OST requirement of ~ 205Å (Mingarro et al., 2000; Whitley et al., 1996; Ismail et al 2012). Additionally, as the P-NST length is increased, if this is due to truncation at different mRNA codons (which I do not know; please see comment 2, above), then test sequences would be in different tunnel locations. Regardless, the following issues should be considered: How many test-sequence residues are helical, which ones are helical, where do they reside, and at which tunnel location does helix formation occur?

Full GpA TM sequence (23 residues) was included in our construct (Supplementary Fig. 1). We do not know the precise number of residues in helical conformation and agree with the reviewer's estimations. To bridge the distance between the glycosylation acceptor site and the PTC the nascent chains probably do not need to harbour a full-length (20-23 residues) helix. Nevertheless, at least 14 residues appear to be needed according to the new data included in Supplementary Fig. 9 (page 14, line 331-334).

4). Figure 1a, right cartoon should be drawn more accurately, e.g., the helix is shown spanning all 100Å of the tunnel and no downstream Lep residues, which we assume are extended. Given comments 2 and 3 above, perhaps there is a better way to represent the relative distances of the green test sequence, the putative α -helix region, and the Lep fragments in the 100Å-tunnel.

We have now modified the cartoons depicted in Fig. 1a in order to clarify this point. Nevertheless, ribosome cartoon is not drawn to scale to respect to the length of the nascent polypeptide chain. We have included this fact in the Figure legend (page 20).

5). The calculations and explanations for Supplementary Figure 5, and the accompanying text in the Results and Methods, need to be expanded and explained. It is not clear to the reader how the residue distribution along the tunnel (Δ PTC) was determined, nor what assumptions were made. See further comments/questions below under Supplementary Figure 5.

We agree with the reviewer that this figure was overly complicated and have removed it from the Supplementary Material. The analysis was originally performed using only the residues present in our simulations; however, this was not an exhaustive sampling of interactions between these residues and the bacterial ribosome. Instead, we have referred to previous work by Pande and coworkers², which mapped out complete free energy profiles for specific side chains within an archaeal ribosome (see page 8 of the revised manuscript).

Specific comments

Title: The title may be overstated. What does “Far” mean, especially given the distance issues in General comment 3 (above) and the specific comments below? “Far” depends on how many of the test residues form a helix, where in the tunnel this helix forms, and where non-TM test sequences eventually compact. Stating “...Far Inside the Ribosome Tunnel...” suggests that a positive control, i.e., soluble helices in the distal regions nearer to the exit port do form a helix. But, it is not clear that this control experiment was done. Perhaps a title like “Intrinsic ability of peptide chains to attain a compact structure within the ribosome” would be more appropriate.

We appreciate referee’s comment but we consider the title to reflect the findings in the manuscript. Regarding the folding of soluble helices in distal regions nearer the exit zone of the tunnel, please see our response to the following Specific comment.

Line 87: “non-TM stretches do not attain full helical conformations inside the ribosome.” This may be worded a little too strongly since there is evidence in the literature for non-TM sequences being compact in the ribosome, and---see above comment---depends on location. Or do the authors want to imply that the soluble helical test sequences do not fold anywhere along the 100Å-length?

In line with the title and the rest of the manuscript we have tuned down this sentence by adding ‘far’ (see main text page 4, line 89) to emphasize the fact that in our constructs the tested helices are close to the ribosome PTC site.

In order to test if the soluble helical test sequences (NAGK and L9) adopt a more compact conformation when placed away from the PTC site, we have engineered two new constructs in which we have moved the glycosylation acceptor site towards the C-terminal end, as follows:

NAGK upstream (for d=67 -16 aa from PTC) construct:

...CEQSTGVTYSN¹STSDFVQTFSTRNGGEATSGFFEVP^{MIS}RDDAAQVAKVLSEALPYI
RRFVHMRLSERKETLGDVTH⁶⁷↓RILTV⁷³...

rib. L9 upstream (for d=67 -16 aa from PTC) construct:

...CEQSTGVTYSN¹STSDFVQTFSTRNGGEATSGFFEVP^{MIK}ALEAQKQKEQRQAEEELA
NAKKHMRLSERKETLGDVTH⁶⁷↓RILTV⁷³...

We include for reviewing purposes a representative SDS-PAGE autoradiography image plus a bar representation of three independent experiments for the translation/glycosylation products of these constructs together with the appropriate controls with 7 residues between the end of the putative helical sequence and the PTC site for truncates with d=67 (-7 aas from PTC) shown in the following Figure (sequences on top, see also Supplementary Fig. 1). The glycosylation level observed in this caption implies that the soluble test sequences are unable to adopt a compact conformation when placed 16 residues away from the PTC (see cartoon at the right). We have included the glycosylation level (dashed lines, with standard deviation as transparent boxes)

for nascent chains with $d=67$ (-7 aas from PTC) harbouring gp41 (green) and VSV-G (blue) TM sequences.

Line 118: “ $\sim 3.4\text{\AA}$ ”. An estimate of $3.2\text{\AA}/\text{aa}$ has been used to analyze pulling forces during translocon-mediated membrane integration by these authors in a previous study (Ismail et al. 2012) and a total distance from PTC to OST (P-NST) was determined to be 205\AA .

We think that $\sim 3.4\text{\AA}$ is the more common value used for a fully extended peptide conformation in the field³⁻⁶.

Lines 133-136: As cited in the above comments, it is not clear which part of the test sequence forms a helix and where in the tunnel this happens, e.g., could it be that in the constriction region the C-terminus of the test TM sequence is extended, but has its N-terminal region compact below the constriction?

We do not know which part of the hydrophobic test sequences forms a helix and where in the tunnel this happens precisely, but it has been observed previously that a nascent TM segment folds into a compact conformation as soon as is located only 5-6 residues away from the PTC site⁷, in good agreement with our constructs location. This work is now referred on page 12.

Lines 174-177: How do the implied stabilizing effects (“higher affinities”) between peptide side-chains and the tunnel at different regions from the PTC, shown in Supplementary Fig. 5, compare with Pande’s free-energy profiles at different tunnel locations (Petrone et al., 2008)?

We thank the reviewer for reminding us of this paper. Given the previously reported affinities for amino acids in Petrone *et al.* and the lack of exhaustive sampling from our work we have removed the analysis of our simulations and

refer instead to the cited paper² for this analysis. We find that their identification of a free energy barrier for large, hydrophobic side chains near the exit pore is consistent with our findings that these residues are stabilized when they are within the tunnel in the helical conformation and not exposed to solvent. We have updated the text accordingly on page 8 to address this question.

Lines 219-220: Regarding the 0.5 TM glycosylation experiments (Fig. 5), the 0.5 segments are the N-terminal residues of the full-length VSV-G or gp41. Where are these 0.5 segments inserted in the Lep peptide? Are they in the identical peptide locations as they are in the full-length peptide, with the deleted C-terminal test residues replaced by additional 10-11 residues from Lep (made by using a new restriction site in the Lep mRNA) that are downstream of the 0.5 segment? Alternatively, is the mRNA restriction site unchanged but the C-terminal 10-11 residues are mutated to be extended? Regardless, what is the sequence of these replacement residues, can they influence folding, and are the N-terminal 0.5 residues in a different tunnel location due to the extended downstream 10-11 Lep residues? This information and clarification should be included.

We used the N-terminal half of VSV-G TM helix because it was the part studied previously by FRET experiments⁴. The sequence of replacement residues (from Lep P2 domain) is shown in Supplementary Fig. 1 and correspond to a non-helical region in the P2 domain structure (Supplementary Fig. 2c). Truncated nascent chains were obtained by generating truncated mRNAs at the appropriate codon location. See also our response to General comment 2 and our response to point (1) Reviewer #3.

Line 290: “our current study in which the start of the test sequences is located ~ 8 residues from the P-site”. Please clarify if this is only for the 67 P-NST length peptide or for all full-length test constructs, depending on how the constructs were made (see General comment 2), above).

The positions are similar for all full-length test constructs. The helices are located 7-9 residues (due to helical length differences) away from the PTC for all truncated nascent chains with P-NST d=67. Shorter and longer nascent chains reduced and increased the number of residues in this region, respectively (see Supplementary Fig. 1). We hope we have clarified this issue in the body text (page 12-13) and with new Supp. Fig. 1.

Line 291: Given the above concern about location and number of residues in the helix (General comment 2), it might be better to avoid stating that adoption of a helical structure is initiated at the P-site.

As stated (page 13), we think that the adoption of α -helical conformation occurs ‘near’ the P-site.

Line 472: Typographical error: should be Supplementary Fig. 6.

This error has been corrected.

Figure 1: Please see General comments 2-4, above, regarding Fig. 1a. In Fig. 1b, why is there an apparent glycosylation band in lane 1 in the absence of RM (no OST; compare lanes 1 and 2)? Is this a background band and if so, does it vary with P-NST length, or is it constant and does not contribute to the intensity of the glycosylation band?

A faint band could be appreciated in some gels, but when it appears it is constant for the different lengths in the same construct and does not contribute to the quantification of the glycosylation bands.

Figure 5: Please see above comment, Lines 219-220. In addition, could the authors comment on why there is a small protected singly-glycosylated H2/inserted fragment, i.e., upward single black triangle, lane 6 in Fig. 5c and d? Does this suggest some fraction of 0.5 TM inserts?

Yes, in both constructs we observed some fraction of TM.5 inserted (see lanes 5 in both gels). In fact, the level of insertion is higher for gp41 TM.5 (81%) than for VSV-G TM.5 (31%), and this is the reason why the band highlighted by upward single black triangle is more evident in Fig. 5d (lane 6) than in Fig. 5c (lane 6). To clarify this point we have now included the experimental values obtained for the membrane insertion of these two sequences in the body text (page 11).

Supplementary Figure 2: What length peptide is used here for each test peptide? 73? 67? This should be consistent with the % glycosylation in lanes 2, 5, 8. In addition to unglycosylated and singly-glycosylated bands, why is there a top band in lane 2 (VSV-G, +RM), whereas lane 1 (VSV-G, -RM) only shows full-length TM peptide? Does this suggest a doubly-glycosylated species is present in lane 2?

These are full-length translation products, including full P2 domain from Lep at the C-terminus with a theoretical molecular weight of 34.9 kDa (see new Supplementary Fig. 3b, uncropped gel). The top band in lane 2 (VSV-G, +RM) does not correspond precisely with the mobility observed for doubly glycosylated molecules as shown in the corresponding uncropped gel mentioned above, but instead it is a product from non-related mRNAs probably contaminating the translation/glycosylation assay ingredients.

Supplementary Figure 3: This approach is quite good and is the basis for the results shown in Fig. 2. With regard to Supplementary Fig. 3, Δ SASA (bottom panel), if as stated above (General comment 3 and Lines 133-136), only some of the VSV-G and gp41 residues (solid lines) are helical and their location with respect to tunnel interaction sites matters, then I would expect the differential $\Delta\Delta$ SASA curves for VSV-G and gp41 versus NAGK and L9 to be less dramatic.

Nonetheless, the simulations point out one extreme endpoint, but how does this relate to the glycosylation results and the above consideration?

We have tracked the secondary structure of helical simulations and found that for all the sequences, while some secondary structure is lost during the simulation, at least half of the sequence does remain helical during the entire simulation as shown in the next Figure:

Secondary structure of nascent peptides within the ribosomal exit tunnel. Secondary structure of residues 38 to 60 (58 for VSV-G) of the nascent polypeptides in the helical conformation within the ribosomal exit tunnel during 200ns of equilibrium molecular dynamics simulations. Secondary structure codes: (T) turn, (E) extended, (B) isolated bridge, (H) α -helix, (G) 3_{10} -helix, (I) π -helix, and (C) coil (random or unstructured)

Critically, the portions that lose their helical structure still remain compact and are close to a helical conformation (Supplementary Figure 5).

Supplementary Figure 5: This Figure, and the accompanying text in the Results and Methods, are not adequately explained or defined. For example, is this the 67-residue peptide? What sequences are used in each panel? How is the "Distance from PTC (Å)" calculated, since this depends on the conformation and location of each residue? For each type of residue evaluated, does this represent an in silico mutagenesis of each residue in each peptide? Or a

frequency distribution for each type residue in each native test sequence? How were each of the 4 test peptides analyzed to give the distributions shown in Supplementary Fig. 5? Please define the color-coding. Finally, how do the results shown in Supplementary Fig. 5 relate to the functional glycosylation results and implications for compaction, and directly to the test peptide sequences? The above issues need to be explicitly stated in the Results, Methods, Legend, and perhaps the Discussion.

We agree completely with the reviewer that this figure was overly complicated. We removed it and the accompanying analysis from the manuscript. Instead we have referred to previous work by Pande and coworkers², who mapped out complete free energy profiles for specific side chains within an archaeal ribosome (see page 8 of the revised manuscript). See also our response to a previous question above.

Reviewer #3

General comments

(1) Is there a critical length for helix formation? In other words, if length is key what should the minimum length be for TM helix?

We have prepared new constructs by deleting 3 residues (~one helical turn) at a time from the full VSV-G TM segment, then generating three new constructs with 17, 14 and 11 hydrophobic residues. *In vitro* translations in the presence of microsomes of mRNA truncated samples with d=67 of these new constructs together with the previous full TM and TM.5 constructs [10 (TM.5), 11, 14, 17 and 20 (full TM) hydrophobic residues] showed distinct glycosylated protein bands for 10 and 11 hydrophobic residues (see Supplementary Fig. 8a), suggesting that 10 or 11 residues are not sufficient to be helical in the ribosome tunnel. Interestingly, these values nicely correlate with the insertion capacity of these sequences when assayed in the full length Lep system, in which constructs with 14, 17 and 20 hydrophobic residues are fully membrane inserted via the Sec61 translocon (Supplementary Fig. 8b). We have incorporated these data in the body text (page 14).

(2) It is unclear to me why the authors argue that hydrophilic sequences do not form helices in the tunnel. As I recall, Beckmann (not just ref. 44) found that many sequences (not necessarily TM only) can form compact helices although the extent of helix formation might vary depending on the location in the tunnel. Are the results here not consistent with these older findings?

Beckmann's lab (reference 11) analysed by cryoEM the folding of an alanine-based designed sequence [GGA(EAAAK)₅AGG] inspired by a short peptide that has a strong propensity to form a hydrophilic α -helix in solution⁸. As Reviewer #2 pointed out, the authors observed density consistent with α -helix formation in two different locations in the tunnel. In the cryoEM data Beckmann and colleagues visualized density that can account for four of the five EAAAK

repeats adopting an α -helical conformation when placed 19 residues away from the PTC (-19), but a more extended conformation when the designed sequence was placed only 7 residues away from the PTC (-7). We would like to stress that the later location is precisely the equivalent to our test soluble helices for nascent chains with $d=67$ residues (P-NST) (see Supplementary Fig. 1), where we observed a glycosylation pattern compatible with an extended conformation. Probably, the differences in sequence length (27 residues in the alanine-based peptides vs 22 residues in our soluble helices) and the salt-bridge network designed to promote helical conformation in the alanine-based peptides can afford for the differences observed.

Nevertheless, we have analysed the folding of our soluble helices in a more distal location from the PTC (-16) and found a glycosylation level fully compatible with a non-compact conformation (see above our response to Specific comment line 87 from Reviewer #2).

(3) A mechanism suggested by the authors is that there are favorable intramolecular interaction in the hydrophobic TM sequences. It has been shown theoretically (Ziv PNAS vol 102, p18956) that confinement (hence entropy considerations) could stabilize helices. Why this not hold here? In some papers by Deutsch this notion was invoked.

We have now included the entropic stabilization hypothesis and cited the original work appropriately (pages 13 and 14).

(4) In the above cited paper the authors showed that length of the polypeptide chain plays an important role as surmised in this article. However, the mechanism appears to be different.

As stated in the previous point, we have included that under confinement longer sequences are more likely to adopt helical conformation than shorter ones (page 14).

I am not clear about the MD simulations. How can the authors assure me that they have sampled the conformations of the confined chain to draw statistically reliable conclusions about the equilibrium between helix and unfolded structures? This pertains to Fig. 2 and others in the SI.

We agree that the space may not be fully sampled. We extended our new simulations from 100 ns to 200 ns. Although likely still insufficient, we note that in two distinct structures of translating ribosomes, one mammalian and one bacterial, the nascent peptides are in nearly identical conformations (average distance between $C\alpha$ s of 1.65 Å), suggesting that the accessible conformational space is relatively limited. We have added the following text on page 7:

Although the conformational space of the peptide is unlikely to have been fully sampled in 200 ns, the initial position of the nascent polypeptide was based on that already present in the cryo-EM structure²⁶ (see Methods), which is similar to that of a nascent peptide in a translating ribosome²⁷.

Additionally, with the exception of the TM VSV-G sequence, we observed that the SASA converges in both the helical and extended conformations (Supplementary Fig. 4). For the VSV-G sequence, fluctuations may be the result of a significant loss of helical content in the helical conformation, with only ~6 out of 20 residues remaining helical, albeit still in a compact, near-helical conformation, by the end of the 200 ns simulations (see the Figure included in our response to Specific comment to Supplementary Fig. 3 from Reviewer #2).

There appears to be an inverse correlation between glycosylation efficiency and compaction versus extension. It would be nice get a quantitative relationship, and in turn with the extent of helix formation. Because I am unclear about the MD results this is important for the readers. Some quantitative connection between the major experiments and helix formation is needed.

In order to visualize the quantitative connection between the compactness of the nascent polypeptides (translation/glycosylation data) with the helical state stabilization (MD simulations data), we have plotted the $\Delta\Delta\text{SASA}$ averaged values over the final 100 ns of the simulations *versus* the glycosylation (%) values for nascent chains with distance 67 ($d=67$) between the glycosylation site and the P-site at the ribosome.

As can be observed, there is a strong correlation between the low glycosylation efficiency and helical stabilization for the nascent polypeptides harbouring the TM sequences (gp41 and VSV-G), whilst higher glycosylation efficiency values correlate with less helical conformations ($\Delta\Delta\text{SASA}$ values close to 0) for soluble (non-TM) helical sequences (NAGK and ribosomal L9).

Bibliography

- 1 Behrmann, E. *et al.* Structural snapshots of actively translating human ribosomes. *Cell* **161**, 845-857, doi:10.1016/j.cell.2015.03.052 (2015).
- 2 Petrone, P. M., Snow, C. D., Lucent, D. & Pande, V. S. Side-chain recognition and gating in the ribosome exit tunnel. *Proc Natl Acad Sci U S A* **105**, 16549-16554, doi:10.1073/pnas.0801795105 (2008).
- 3 Lu, J. & Deutsch, C. Secondary structure formation of a transmembrane segment in Kv channels. *Biochemistry* **44**, 8230-8243, doi:10.1021/bi050372q (2005).
- 4 Woolhead, C. A., McCormick, P. J. & Johnson, A. E. Nascent membrane and secretory proteins differ in FRET-detected folding far inside the ribosome and in their exposure to ribosomal proteins. *Cell* **116**, 725-736 (2004).
- 5 Bhushan, S. *et al.* alpha-Helical nascent polypeptide chains visualized within distinct regions of the ribosomal exit tunnel. *Nat Struct Mol Biol* **17**, 313-317, doi:nsmb.1756 [pii] 10.1038/nsmb.1756 (2010).
- 6 Holtkamp, W. *et al.* Cotranslational protein folding on the ribosome monitored in real time. *Science* **350**, 1104-1107, doi:10.1126/science.aad0344 (2015).
- 7 Lin, P. J., Jongsma, C. G., Pool, M. R. & Johnson, A. E. Polytropic membrane protein folding at L17 in the ribosome tunnel initiates cyclical changes at the translocon. *J Cell Biol* **195**, 55-70, doi:10.1083/jcb.201103118 (2011).
- 8 Marqusee, S. & Baldwin, R. L. Helix stabilization by Glu-...Lys+ salt bridges in short peptides of de novo design. *Proc Natl Acad Sci U S A* **84**, 8898-8902 (1987).

Reviewers' Comments:

Reviewer #1:

Remarks to the Author:

The revised manuscript takes care of my previous comments and may be accepted.

Reviewer #2:

Remarks to the Author:

The revision of the manuscript is substantial and for the most part addresses the comments/criticisms. However, despite the excellent new additions and revisions, there are still outstanding issues that should be corrected.

Original General Comments:

For original comments #1 and #2, the missing information was supplied, both in Supplementary Fig. 1. However, part of item #2 is still not clear from the author's answer: If the "C-terminal half of each TM helix was replaced by non-helical residues from Lep P2 domain..." doesn't this change the position of the N-terminal 0.5 TM in the tunnel, i.e., replacing a helical portion with an extended portion will increase the chain length by ~2.3-fold (depends on choice of 3.4Å per residue) and shift the 0.5 TM to a new location? This might impair folding. As stated in my original review, "Did you use a different truncation site in the DNA template to 'compensate' (shorten), so that the N-terminal 0.5 TM segment remains at the same tunnel location?"

Page 14: In the first paragraph, does helical conformation matter to the sequence's alignment with ribosome proteins' hydrophobic amino acids to make contact?

General comment #3: The question was not fully addressed, i.e., "...the following issues should be considered: How many test-sequence residues are helical" It is all right that the authors don't know, but they should perhaps give a more explicit answer in the paper (e.g., on page 14, lines 331-334), perhaps something like "Although we do not know the precise number of residues in helical conformation, we estimate that at least 14 residues appear to be needed (Supp. Fig 9), that they form early, near the PTC and maintain their helicity...into the membrane."

Original Specific Comments:

Title: The title is still somewhat ambiguous and "Far" is a subjective term, which depends on "how many of the test residues form a helix, where in the tunnel this helix forms, and where non-TM test sequences eventually compact." As the authors themselves state in response to the original General Comment #3, "We do not know the precise number of residues in helical conformation and agree with the reviewer's estimations." This means that the authors cannot know or say where the helix is located, "far" into the tunnel or otherwise.

Line 87: Excellent new experiments with relocated glycosylation sites for NAGK and L9.

219-220: The authors have answered one part of the original comment well, i.e., the replacement residues sequence and their extended property. However, they still need to address the important issue in this maneuver, which was originally asked: "...are the N-terminal 0.5 TM residues in a different tunnel location due to extended downstream 10-11 Lep residues?" At the least, the authors could state "Yes, but this doesn't matter since helices are permissible along the whole length of the tunnel⁴⁴ and we hypothesize that hydrophobic helices could maintain their helicity throughout translation into the translocon and integration into the membrane."

291: Given the authors own admission "We do not know which part of the hydrophobic test

sequences forms a helix and where in the tunnel this happens precisely..." it may not be so 'near' the PTC. If, as the authors state, the helices are located 7-9 residues away from the PTC ($d=67$) and we use 3.4\AA per residue, then the distance from the PTC is $\sim 27\text{\AA}$. This is the closest a helix gets to the PTC. If only the more N-terminal upstream segment of the test sequence forms the helix with C-terminal extended, then this distance will be even greater. Neither is "near" (a subjective term) the PTC.

Excellent answer and added information for the Figure 5 Comment.

The new figure shown in response to Specific Comment "Supplementary Figure 3" is somewhat concerning. The authors, in their response, say "We have tracked the secondary structure of helical simulations..." This new figure shows, for early times (before loss of some helicity), according to the color code, that $\sim 50-60\%$ of residues 38-60 for NAGK and L9, which are non-TM, are helical in the tunnel. This value is similar to VSV-G and half as much as for gp41. Why is this? The paper concludes "that folding in the ribosome is attained for TM helices but not for soluble helices." Isn't this new figure contradictory to these conclusions from glycosylation assays? Also, based on the y-axis, don't these simulations suggest that only 11 residues of VSV-G form a helix early in the simulation?

Reviewer #3:

Remarks to the Author:

The authors have done due diligence and answered my questions satisfactorily. I therefore recommend publication.

We thank the reviewers for their very supportive comments. Below, we address the specific points raised by the Reviewer#2 (in blue) and elaborate on the corresponding changes in the manuscript.

Reviewer #2

Original General Comments:

For original comments #1 and #2, the missing information was supplied, both in Supplementary Fig. 1. However, part of item #2 is still not clear from the author's answer: If the "C-terminal half of each TM helix was replaced by non-helical residues from Lep P2 domain..." doesn't this change the position of the N-terminal 0.5 TM in the tunnel, i.e., replacing a helical portion with an extended portion will increase the chain length by ~2.3-fold (depends on choice of 3.4Å per residue) and shift the 0.5 TM to a new location? This might impair folding. As stated in my original review, "Did you use a different truncation site in the DNA template to 'compensate' (shorten), so that the N-terminal 0.5 TM segment remains at the same tunnel location?"

In order to investigate the folding of half TM segments (TM.5) we decided to focus on the N-terminal (Nt) half to compare our data with similar experiments (FRET) previously published (ref. 15). Then, the deleted C-terminal (Ct) region was replaced by an expected extended region from our model protein (Lep). As a consequence, if this segment (see the scheme below, replacement) adopts an extended conformation the upstream N-t half of the TM segment would indeed be located farther away from the P-site, as the reviewer pointed out, but with precisely the same number of amino acid residues bridging the VSV-G segment to the P-site (see Supplementary Figure 1). Using the same truncation site in the DNA template carrying the C-terminal half deletion will also modify the Nt half location, this time by placing it closer to the P-site (Ct-deleted). Alternatively, we could delete the N-terminal instead of the C-terminal half, with the C-terminal half remaining at the same position (Nt-deleted) but we would lose the possibility to compare our data with that published on ref. 15.

We think that the replacement strategy that we have used in our work does not compromise the folding capacity of the nascent polypeptide too much since replacing 10 residues in a helical conformation ($1.5\text{\AA} \times 10$) by 10 residues in an extended conformation ($3.4\text{\AA} \times 10$) would result in $\sim 19\text{\AA}$ displacement away from the P-site, which still will locate the TM.5 region in the proximity of the constriction point of the tunnel where helical stabilization will be most

pronounced (see also Supplementary Figure 5a as well as our responses to the following comment and to lines 219-220 comment). Nevertheless, we have highlighted this concern in the manuscript main text (page 10, lines 232-234).

Page 14: In the first paragraph, does helical conformation matter to the sequence's alignment with ribosome proteins' hydrophobic amino acids to make contact?

As can be seen in Supplemental Figures S5a and S8, most of the loss of helical content for VSV-G, for example, occurs in the N-terminal residues, which are outside the constriction region between ribosomal proteins L4 and L17 and in a less hydrophobic environment. Therefore, the placement of the hydrophobic residues in this region will be less important than those near the constriction region. Helical stabilization by the ribosome will be the most pronounced within the constriction region. And even though the peptides lose some of their helical character, they remain compact, in approximately the same placement, meaning the hydrophobic residues of the nascent peptide, particularly at the constriction point where the peptides stay helical, stay in roughly the same positions during the entire simulation. In summary, our results about hydrophobic contacts should not depend too much on the observed loss of helicity or the sequence's alignment with ribosome proteins' hydrophobic amino acids to make contact.

General comment #3: The question was not fully addressed, i.e., "...the following issues should be considered: How many test-sequence residues are helical" It is all right that the authors don't know, but they should perhaps give a more explicit answer in the paper (e.g., on page 14, lines 331-334), perhaps something like "Although we do not know the precise number of residues in helical conformation, we estimate that at least 14 residues appear to be needed (Supp. Fig 9), that they form early, near the PTC and maintain their helicity...into the membrane."

We agree with the reviewer's comment and have included this suggestion in the main text (page 15, first paragraph).

Original Specific Comments:

Title: The title is still somewhat ambiguous and "Far" is a subjective term, which depends on "how many of the test residues form a helix, where in the tunnel this helix forms, and where non-TM test sequences eventually compact." As the authors themselves state in response to the original General Comment #3, "We do not know the precise number of residues in helical conformation and agree with the reviewer's estimations." This means that the authors cannot know or say where the helix is located, "far" into the tunnel or otherwise.

We agree with the reviewer's comment and removed the subjective term 'far' from the title.

Line 87: Excellent new experiments with relocated glycosylation sites for NAGK and L9.

Thanks a lot for the supportive comment.

219-220: The authors have answered one part of the original comment well, i.e., the replacement residues sequence and their extended property. However, they still need to address the important issue in this maneuver, which was originally asked: "...are the N-terminal 0.5 TM residues in a different tunnel location due to extended downstream 10-11 Lep residues?" At the least, the authors could state "Yes, but this doesn't matter since helices are permissible along the whole length of the tunnel⁴⁴ and we hypothesize that hydrophobic helices could maintain their helicity throughout translation into the translocon and integration into the membrane."

The observation made by the reviewer is correct: the Nt half of each TM segment is placed in a different tunnel location when the Ct half is replaced by an expected extended segment. However, we think that this is not an insurmountable inconvenience to investigate the folding of these nascent chains because as reported before (and pointed out by the reviewer) helices are permissible along the whole length of the tunnel (ref. 44). This issue is explicitly mentioned on page 14 (lines 332-337), and related to our response to the first issue of this second revision.

291: Given the authors own admission "We do not know which part of the hydrophobic test sequences forms a helix and where in the tunnel this happens precisely..." it may not be so 'near' the PTC. If, as the authors state, the helices are located 7-9 residues away from the PTC ($d=67$) and we use 3.4\AA per residue, then the distance from the PTC is $\sim 27\text{\AA}$. This is the closest a helix gets to the PTC. If only the more N-terminal upstream segment of the test sequence forms the helix with C-terminal extended, then this distance will be even greater. Neither is "near" (a subjective term) the PTC.

Again, we agree with the observation made by the reviewer. We just wanted to emphasize in our work that the folding is not occurring in the vestibule of the tunnel but closer to the PTC. To avoid misleading wording we have removed the term "near" and replaced by "20-30 \AA away from" (page 13, line 306).

Excellent answer and added information for the Figure 5 Comment.

Thank you very much for the supportive comment.

The new figure shown in response to Specific Comment "Supplementary Figure 3" is somewhat concerning. The authors, in their response, say "We have tracked the secondary structure of helical simulations..." This new figure shows, for early times (before loss of some helicity), according to the color code, that $\sim 50-60\%$ of residues 38-60 for NAGK and L9, which are non-TM, are helical in

the tunnel. This value is similar to VSV-G and half as much as for gp41. Why is this? The paper concludes “that folding in the ribosome is attained for TM helices but not for soluble helices.” Isn’t this new figure contradictory to these conclusions from glycosylation assays? Also, based on the y-axis, don’t these simulations suggest that only 11 residues of VSV-G form a helix early in the simulation?

We realized that the strict, binary measure of helicity used in the figure is insufficient to capture the variety of near- α -helical and other compact states observed in our simulations. Therefore, we have added one additional supplementary figure, Fig. S9, in which α -helical content is tracked as a continuous variable between 0 and 1 over time (panel A). Additionally, the compactness (panel B) is tracked. While the α -helical content of VSV-G drops to as low as 0.6, the end-to-end distance increases by less than 15%, demonstrates that it remains relatively compact, and in good agreement with glycosylation assays.

Supplementary Figure 9: Compactness of the TM segments VSV-G and gp41 in simulations in the mammalian ribosome PET. In all panels, lighter colors are used for the folded state and darker colors for the fully extended state. (A) α -helical content as measured in NAMD. Briefly, this collective variable takes as input both the angle between every three successive C_{α} atoms and the strength of hydrogen bonds between backbone N and O atoms of every i and $i+4$ residues; 1.0 is a perfect α -helix and 0.0 is no helical content whatsoever. (B) End-to-end distance over time of the helical residues, i.e., residues 38-58 for VSV-G and 38 to 60 for gp41. (C-D) Distance from the PTC to the (C) C-terminus and to the (D) N-terminus.

We have added a sentence to the main text (page 12, lines 266-269) to address this concern.

To summarize, each system was built with the peptide in a compact (helical) state or in an extended state, which does not change appreciably over the course of the simulations. Nonetheless, the simulations allowed us to calculate the degree of ribosome-peptide hydrophobic interactions, which suggest that each peptide has a preferred state, whether compact (gp41 and VSV-G) or extended (NAGK and L9). We expect that over orders-of-magnitude longer time scales, all systems would relax to their preferred states.

Reviewers' Comments:

Reviewer #2:

Remarks to the Author:

The Bano-Polo manuscript is much improved and should be accepted for publication. I thank the authors for their really thoughtful efforts and new analyses in the revisions in response to feedback.

1). I acknowledge the rationale for using the replacement strategy. Although it is an assumption that a 19Å-displacement from the P-site (significant and peptide does interact with different ribosome components) does not compromise the folding capacity, the authors did highlight this issue and appropriately alert the reader.

2). Similarly, the authors qualify appropriately their knowledge of number of residues in helical conformation.

3). Title change is now more accurate without losing any impact of the message. Similarly, with replacement of "near" with "20-30Å away from".

4). New Supplementary Figure 9 and the added commentary are fine and make a more compelling case for the authors' conclusions.